# The Immobilization of an FGF2-Derived Peptide on Culture Plates Improves the Production and Therapeutic Potential of Extracellular Vesicles from Wharton’s Jelly Mesenchymal Stem Cells

**DOI:** 10.3390/ijms251910709

**Published:** 2024-10-04

**Authors:** Youngseo Lee, Kyung-Min Lim, Hanbit Bong, Soo-Bin Lee, Tak-Il Jeon, Su-Yeon Lee, Hee-Sung Park, Ji-Young Kim, Kwonwoo Song, Geun-Ho Kang, Se-Jong Kim, Myeongjin Song, Ssang-Goo Cho

**Affiliations:** 1Department of Stem Cell and Regenerative Biotechnology, Molecular & Cellular Reprogramming Center and Institute of Advanced Regenerative Science, Konkuk University, Seoul 05029, Republic of Korea; bbeock0621@naver.com (Y.L.); lmin0217@naver.com (K.-M.L.);; 2R&D Team, StemExOne Co., Ltd., 307, KU Technology Innovation Building, 120 Neungdong-ro, Gwangjin-gu, Seoul 05029, Republic of Korea; 3New Materials R&D Center of AMOGREANTECH Co., Ltd., 609-1 Wolha-ro, Haseong-myeon, Gimpo-si 10011, Republic of Korea

**Keywords:** exosome, fibroblast growth factor (FGF2), peptide, Wharton’s jelly mesenchymal stem cell (WJ MSC), wound healing, anti-inflammation

## Abstract

The skin is an essential organ that protects the body from external aggressions; therefore, damage from various wounds can significantly impair its function, and effective methods for regenerating and restoring its barrier function are crucial. This study aimed to mass-produce wound-healing exosomes using a fragment of the fibroblast growth factor 2 (FGF2)-derived peptide (FP2) to enhance cell proliferation and exosome production. Our experiments demonstrated increased cell proliferation when Wharton’s jelly mesenchymal stem cells (WJ MSCs) were coated with FP2. Exosomes from FP2-coated WJ MSCs were analyzed using nanoparticle-tracking analysis, transmission electron microscopy, and Western blotting. Subsequently, fibroblasts were treated with these exosomes, and their viability and migration effects were compared. Anti-inflammatory effects were also evaluated by inducing pro-inflammatory factors in RAW264.7 cells. The treatment of fibroblasts with FP2-coated WJ MSC-derived exosomes (FP2-exo) increased the expression of FGF2, confirming their wound-healing effect in vivo. Overall, the results of this study highlight the significant impact of FP2 on the proliferation of WJ MSCs and the anti-inflammatory and wound-healing effects of exosomes, suggesting potential applications beyond wound healing.

## 1. Introduction

Wound healing is a complex process involving blood clotting, inflammation, new tissue formation, and tissue remodeling [1,2]. Traditional wound repair methods are based on physical therapy, including dressing changes, wound drainage, and skin grafting. However, traditional therapies are infamous for their long duration and susceptibility to infection [3]. Therefore, stem cell therapy has attracted considerable attention as a new treatment method that complements traditional regenerative medicine.

Mesenchymal stem cells (MSCs), a group of cells with self-renewal activity, have shown substantial developmental potential and have received increasing interest in the field of wound repair [3,4] as they are expected to transform the treatment of refractory wounds [5]. However, there are limitations to treatments using cells themselves, as they pose an increased risk of immune rejection and tumorigenesis [6,7]. Owing to the limitations of these cell therapies, many studies have been conducted on the replacement of Wharton’s jelly mesenchymal stem cells (WJ MSCs) with exosomes. Exosomes are nanoparticles with a 50–200 nm diameter secreted by eukaryotic cells and are heterogeneous bilayer membrane vesicles responsible for cell-to-cell communication. They consist of lipids [8], proteins [9], and nucleic acids [10,11] from their parent cells. EVs have been demonstrated to be an effective drug delivery strategy in multiple areas, including tumors, the spinal cord, the cardiopulmonary system, the brain, and wound surfaces [12,13,14]. Moreover, as a novel cell-free treatment, MSC-derived EV-based therapy has demonstrated good biosafety, high stability, and low immunogenicity. The therapeutic effects of EVs on wound healing and scar repair have been shown to be superior to those of MSC therapy, demonstrating excellent application prospects [15,16].

Therefore, the mass production of exosomes is a crucial research topic for clinical application. However, securing many cells for extracting WJ MSC-derived exosomes may be challenging due to the shortcomings of rapid aging and the high cost of large-scale production [17]. Fibroblast growth factor 2 (FGF2) is one of 22 FGFs and has various functions, such as migration, regeneration, and wound healing; therefore, it is widely used as an adjuvant to promote cell growth in vitro [18]. However, because of its sensitivity to heat and pH, daily supplementation with FGF2 during cell culture is impractical. Although the use of recombinant FGF2 can address this issue, the purification process is expensive, rendering it economically unfeasible [19,20].

In this study, we present a fragment peptide of FGF2 (FP2) that is more economical to produce than recombinant FGF2. Unlike the unstable recombinant FGF2, this peptide can be fixed to the bottom of the cell culture plate, ensuring that it can act on all WJ MSCs grown in a single layer. Previously reported FGF2 fragment peptides, such as canofin1 (FP1), hexafin2 (FP3), and canofin3 (FP4), are known neurotrophic factors; however, the role of FP2 used in this study is not well understood [21,22,23].

FP2 is fused with the C-terminus of mussel adhesive protein (MAP) and immobilized by pre-activating the cell culture dish with an EDC-NHS solution [21]. MAP is recognized for its strong adherence to rough and moist surfaces [24], functioning as a linker connecting the peptide to the bottom of the cell culture dish and cells. Previous studies have shown that WJ MSCs cultured in FP2-coated dishes have a higher proliferation rate than those in uncoated dishes or those coated with other peptides and that they maintain their subculturing capability for a longer period [25].

This study examined the effect of coating with FP2 on the generation of exosomes in WJ MSCs (FP2-exo) and the migration, anti-inflammation, and wound-healing effects of the produced exosomes. 

## 2. Results

### 2.1. Assessment of FGF2 Peptide Variants to Boost Proliferation and Exosome Yields in WJ MSCs

The FGF2 peptide structures were predicted using the PyMOL Viewer (DeLano Scientific LLC, San Francisco, CA, USA) and showed two protrusions rotated at 180° (Figure 1A). Canofin1, FP2, hexafin2, and canofin3 were marked in different colors. The FGFR1-binding site of each peptide was predicted using the CABS-dock program and is illustrated in the figures (Figure 1B). Canofin1, hexafin2, and canofin3 each had one FGFR1-binding site, whereas FP2 had three.

Based on this prediction, four FGF2-derived peptides were screened. The cell culture dish was first coated with EDC/NHS, a zero-length crosslinker, followed by substitution with each peptide fused to MAP. WJ MSCs were grown in these coated dishes, and exosomes were isolated from each peptide-coated dish to compare the amount of exosome production.

The screening results confirmed that cell proliferation was optimal when WJ MSCs were cultured in dishes coated with FP2. Furthermore, when exosomes were isolated from WJ MSCs grown on dishes coated with FP2, the number of particles per cell was approximately twice that of other peptides (Figure 1D). Based on the above results, FP2 was selected for coating, and the wound-healing effect was confirmed after isolating the exosomes from the conditioned medium (Figure 1C).

### 2.2. Investigating Features of Exosomes Produced by FP2

As the FP2 coating resulted in the highest exosome production, three types of exosomes were compared: exosomes isolated from WJ MSCs grown on uncoated plates (CON-exo), plates coated with FP2 (FP2-exo), and plates coated with MAP (MAP-exo). Transmission electron microscopy (TEM) revealed similar sizes and round shapes of all three exosome types (Figure 2A). Nanoparticle-tracking analysis (NTA) confirmed that the sizes of exosomes separated from CON-exo, MAP-exo, and FP2-exo were similar (Figure 2B). The analysis results show that the particle size was in the range of 50–150 nm, indicating that FP2 and MAP did not significantly alter the EV size and that the largest number of particles could be obtained when exosomes were extracted from the dish coated with FP2 fused to MAP, consistent with the results in Figure 1D.

To normalize the exosome samples before Western blot analysis, the gel was stained with 0.05% Coomassie blue G250 (Figure 2C). Western blot analysis was then performed to evaluate the effects of FP2 and MAP on the EV characteristics (Figure 2D). The results show that the essential characteristics of the EVs did not change after coating with FP2. The tetraspanin proteins CD63 and CD9, which are EV-positive markers, were detected in all the CON-exo, MAP-exo, and FP2-exo groups, but the endoplasmic reticulum protein calnexin and Golgi protein GM130, which are known EV-negative markers, were not detected.

Briefly, our results show that the production of EVs was the highest in dishes coated with FP2 and that the characteristics of the EVs, such as EV size and exosome marker expression, were maintained.

### 2.3. Coating with FP2 Elevated Expression Levels of Genes Contributing to Exosome Biogenesis

Because of the approximately two-fold increase in FP2-exo production, we confirmed the RNA expression associated with exosome biogenesis using RT-qPCR. Initially, genes related to the formation of multivesicular bodies, including CD63, CD9, and nSMase, were assessed. CD63 and CD9 are known for their abundance in the exosome membrane as endosome-specific tetraspanins, whereas nSMase induces endocytosis by generating ceramide in the inner lobule of the plasma membrane. We also examined the genes involved in exosome release, such as Rab35 and 27b. CDC42, which is regulated by protein kinase B (AKT) and is downstream of the FGF2-FGFR1 signal, triggers endocytosis through a dynamin-independent process. A comparison of gene expression levels confirmed that exosome biogenesis-related gene expression was significantly increased when FP2 was used for coating compared with cases with no coating or when only MAP was used (Figure 3).

### 2.4. Confirming Migration-Promoting Effect of FP2-exo in Human Fibroblasts

To verify the impact of FP2-exo, we conducted cell viability and migration assays after exosome treatment. Initially, the cells were exposed to exosomes at concentrations of 1 × 10^8^ and 1 × 10^9^ to assess their viability. FP2-exo was confirmed to exert the most significant effect, with no notable differences between CON-exo and MAP-exo (Figure 4A). Subsequently, we confirmed the absence of toxicity when treating the cells with exosomes; furthermore, at a concentration of 1 × 10^9^, the proliferation effect was optimal.

To evaluate cell migration after exosome treatment, Transwell and scratch assays were performed (Figure 4B,C). After 24 h, cells treated with FP2-exo exhibited the highest migration, with no significant difference in the migration area compared with those with CON-exo and MAP-exo. Additionally, the in vitro scratch assays revealed that wound closure was the fastest in cells treated with FP2-exo. These results demonstrate that among CON-exo, MAP-exo, and FP2-exo, FP2-exo exhibited the most effective in vitro wound-healing ability. Furthermore, there was no difference in the wound-healing effects between CON-exo and MAP-exo. This confirmed that the MAP peptide segment attached to the FGF2 peptide did not contribute to the wound-healing effect of WJ MSCs through exosome production.

### 2.5. FP2-exo Efficiently Reduced Expression of Pro-Inflammatory Factors Induced by Lipopolysaccharide (LPS)

RAW 264.7 cells were exposed to LPS to assess the levels of nitric oxide (NO), inducible nitric oxide synthase (iNOS), cyclooxygenase 2 (COX2), and cytokines IL1β and IL6 (Figure 5A,B). The control group was not subjected to LPS treatment, whereas all other groups were treated with LPS to induce the M1 macrophage phenotype. Subsequently, these cells were treated with exosomes and dexamethasone (Dex) to evaluate the expressions of proinflammatory mediators and cytokines. Dex, an anti-inflammatory steroid [26], is known to suppress the expression of various inflammatory genes by inhibiting NF-κB [27], a key transcription factor in inflammation. The analysis revealed that treatment with FP2-exo and Dex resulted in similar inhibition of LPS-induced pro-inflammatory mediators. Moreover, this combination treatment restored the RNA expression levels of pro-inflammatory mediators and cytokines to those observed [26] in cells not exposed to LPS.

### 2.6. FP2-exo Enhanced Healing of Skin Wounds, Acting via FGF2 Signal

FGF2 promotes wound healing by activating endothelial cells and fibroblasts. FP2-exo was found to influence cell growth, survival, and migration (Figure 4). Building on this, we examined how FP2-exo activates FGF2 signaling at the protein level using Western blot analysis (Figure 6A). Analysis of FRS2α and AKT, downstream of FGFR1 and FGF2 where FP2 binds, revealed that FP2 exosomes activate the FGF2 signal more effectively than other exosomes or no treatment.

To validate the efficacy of FP2-exo in wound healing, we made an 8 mm wound on the back of a mouse and injected exosomes (Figure 6B). Histological analysis was performed using H&E and Masson’s trichrome staining (Figure 6C). The results demonstrate that wounds treated with FP2-exo healed significantly faster compared with those treated with CON-exo, as evidenced by enhanced tissue regeneration and collagen synthesis in the stained sections. In summary, the results confirm that FP2 promoted cell proliferation, enhanced exosome production, and generated more functional and practical exosomes for wound healing than conventional exosomes.

## 3. Discussion

In this study, we identified a peptide that enhanced the proliferation of human WJ MSCs and increased the secretion of exosomes that are functional for wound healing. WJ MSCs are known to promote wound healing and angiogenesis, and exosomes derived from WJ MSCs inherit these properties while mitigating the risk of uncontrolled cell differentiation or proliferation [28,29,30]. However, because a single cell typically secretes approximately 1000 exosomes, improving their functionality and increasing their production yield are essential goals [31,32].

Mussels secrete a mix of adhesive proteins that enable them to attach firmly to wet surfaces in marine environments. Mussel adhesive protein (MAP) is less toxic to cells than other adhesive materials and exhibits excellent adhesion to various cell types, including human cells, making it a widely used bioadhesive [24]. When MAP is linked to FP2, it acts as a linker to effectively connect the peptide to cell culture dishes. Our findings confirm that the MAP-FP2 fusion peptide significantly enhanced WJ MSC proliferation (Figure 1D).

Analysis of RNA expression related to exosome biogenesis revealed that FP2-coated WJ MSC-derived exosome (FP2-exo) production was approximately doubled (Figure 1D). CD63 and CD9 are endosome-specific tetraspanins abundant in exosome membranes [33,34], and nSMase generates ceramides in the inner lobe of the plasma membrane, inducing endocytosis [35]. The Rab family is involved in endocytosis and secretion pathways, which are crucial for membrane trafficking and exosome release [36]. Additionally, CDC42, regulated by AKT downstream of FGF2-FGFR1 signaling, induces endocytosis through a dynamin-independent process [37,38]. Gene expression analysis showed that the FP2 coating more than doubled the expression levels compared with those in cells on uncoated or MAP-coated plates (Figure 3). Migration, a key aspect of wound healing, requires epithelial cells to maintain intercellular adhesion while migrating over the wound site to restore the epithelial barrier [39,40,41]. FP2-exo had the most significant effect on cell migration (Figure 4B,C).

When macrophages are treated with LPS, an activation signal is transmitted through Toll-like receptor 4 (TLR4), leading to the secretion of pro-inflammatory mediators [42,43], including NO, which is a critical biological regulator involved in various pathological processes [44,45]. iNOS is active only in activated macrophages and plays a role in the immune system [46,47]. COX2, expressed in inflammatory cells, generates reactive oxygen species (ROS) and prostanoids at inflammatory sites, causing tissue damage [48]. The measurement of pro-inflammatory mediators revealed that FP2-exo significantly reduced their expression levels compared with those with CON-exo, showing no significant difference from the anti-inflammatory steroid Dex (Figure 5). This indicates that FP2-exo effectively reduced LPS-induced inflammatory factors.

FGF2-regulated signaling pathways include AKT, which, when phosphorylated, affects biological activities, such as proliferation and survival, through mTOR and GSK3 [49]. AKT plays a critical role in remodeling, regeneration, and re-epithelialization during tissue repair [50]. In FP2-exo, AKT activation through FGF2 signaling enhanced the wound-healing capabilities compared with those with conventional WJ MSC-derived exosomes (Figure 6). However, further investigations into signaling pathways other than AKT related to wound healing are needed.

While this study provides substantial evidence of FP2-exo’s role in promoting wound healing and tissue regeneration (Figure 6B,C), a notable limitation remains. In exosome-based wound-healing studies, histological analysis commonly includes H&E and Masson’s trichrome staining, often supplemented by IHC to assess specific protein expression related to inflammation and tissue repair [51,52]. Although we included H&E and Masson’s trichrome staining to demonstrate tissue regeneration and collagen deposition, we were unable to incorporate IHC data in this study. The addition of IHC in future studies would offer deeper insights into the underlying cellular mechanisms. This would allow for a more comprehensive evaluation of the therapeutic potential of FP2-exo, complementing the histological evidence presented here.

This study confirmed that FP2 effectively promotes cell proliferation and exosome production. The function of FP2-exo was validated both in vivo and in vitro, demonstrating the biological function of FP2 in WJ MSCs. Therefore, future studies are needed to elucidate the potential applications of FP2-exo in diseases other than wound healing through additional experiments, including comparisons with other growth factor-derived peptides.

## 4. Materials and Methods

### 4.1. Cell Culture

Human WJ MSCs were cultured in alpha minimum essential medium (α-MEM, 12561072, Gibco, Waltham, MA, USA) containing 1% penicillin–streptomycin (P/S, 15140-163, Gibco) and 10% fetal bovine serum (FBS, 16000044, Gibco) at 37 °C with a 5% CO_2_ atmosphere. The experimental protocol for obtaining the WJ MSCs was approved by the Konkuk University Institutional Review Committee (7001355-202010-BR-407).

BJ cells (Fibroblasts, CRL-2522, ATCC) were cultured in Dulbecco’s modified Eagle’s medium–high glucose (DMEM-HG, D6429, Sigma-Aldrich, St. Louis, MO, USA) supplemented with 1% P/S and 10% FBS under the same conditions as those used for the WJ MSCs. At 80% confluency, the cells were dissociated and passaged using the TrypLE Select (1X) reagent (12563029, Gibco) for WJ MSCs or with trypsin–EDTA (0.025%) (25300120, ThermoFisher, Waltham, MA, USA) for BJ cells. The cells were then centrifuged for 3 min at 300 g and seeded at a density of 5000 cells/cm^2^.

RAW 264.7 cells were cultured in DMEM (10569010, Gibco) containing 10% FBS and 1% P/S at 37 °C with a 5% CO_2_ atmosphere. The detachment method for the RAW 264.7 cells was the same as that used for the WJ MSCs.

### 4.2. Peptide Preparation and Coating Method for 2D Culture

To coat the culture plate with MAP, a coupling reaction was performed using 10 mM 1-(3-dimethylaminopropyl)-3-ethylcarbodiimide hydrochloride (EDC, A299, A K Scientific, Union City, CA, USA) and 10 mM N-hydroxysulfosuccinimide sodium salt (NHS, V1233, AK Scientific). The EDC-NHS mixture was prepared by dissolving in 20 mM sodium acetate buffer (pH 6.5). This mixture was then added to the culture plate to ensure that the entire surface was covered and incubated for 30 min at room temperature (RT). After aspirating the EDC-NHS mixture, each peptide (0.05 µg/mL) was added to the plate and incubated at RT for 30 min. The plate was then washed thrice with distilled water and dried.

### 4.3. Protein Structure Prediction

The 3D protein structures of the FGF2-derived peptides were predicted using protein sequences sourced from the RCSBPDB database accessed on 15 August 2024 (http://www.rcsb.org) [53]. The PyMOL 3.0 software (http://www.pymol.org) was used to visualize the protein structures.

### 4.4. Exosome Isolation

WJ MSCs were cultured on both control and coated plates. When the cells reached approximately 70% confluency in a culture dish, they were washed twice with phosphate-buffered saline (PBS, 10010031, Gibco), and the medium was replaced with fresh medium containing 10% exosome-depleted FBS.

The medium was then collected after 48 h. To remove the cells and debris, the collected medium was centrifuged at 2000× *g* for 10 min at 4 °C using an Avanti^®^ J-E Centrifuge (369003, Beckman Coulter, Indianapolis, IN, USA). Next, the medium was centrifuged at 10,000× *g* for 30 min at 4 °C to remove microvesicles. The exosomes were then isolated from the supernatant by ultracentrifugation at 187,000× *g* for 2 h at 4 °C using an Optima^®^ L-90K Ultracentrifuge (365672, Beckman Coulter). The exosome pellets were resuspended in 200 µL of filtered 1× PBS. After isolating the exosomes, their concentration and size distribution were determined via nanoparticle-tracking analysis (NTA) using the TWIN ZetaView^®^ system (PMX-220, Withinstrument, Inning am Ammersee, Germany), according to the manufacturer’s instructions.

### 4.5. TEM

The exosomes were loaded onto a grid (FCF300-Cu, EMS) and incubated for 10 min. The excess solution was then absorbed using 3M paper. The grid was washed with distilled water (3D.W.) and subjected to negative staining with 1% phosphotungstic acid. After staining, the remaining solution was absorbed with 3M paper, and the grid was washed again with 3D.W. The exosomes were then observed using a TEM (JEM-1010, Jeol, Akishima, Tokyo, Japan).

### 4.6. Western Blot Analysis

After homogenizing the cells in RIPA lysis buffer (CBR002, LPS solution, Daejeon, Republic of Korea), they were vortexed and incubated on ice for 3 min, five times. The proteins were separated after centrifugation at 13,000 rpm and 4 °C for 15 min. The proteins from cells and exosomes were quantified according to the manufacturer’s instructions using a BCA kit (23227, Thermo Fisher Scientific). Qualified protein samples were electrophoresed on 12-well gels (4–12% Bis-Tris Plus Gels, NW04122BOX, Invitrogen, Waltham, MA, USA) for 30 min. The gel was stained using 0.05% Coomassie blue G250 (CBC006, LPS Solution). The proteins were then transferred onto a membrane using iblot2 transfer stacks (Invitrogen, IB23001). The membrane was treated overnight at 4 °C with Protein-free 5X General-Block Solution (TLP-115.1G, TransLab, Daejeon, Republic of Korea) and primary antibodies.

The primary antibodies used included CD63 (ab109201, Abcam, Cambridge, UK), CD9 (ab263023, Abcam) for an exosome-positive marker, GM130 (12480, CST), calnexin (sc-23954, Santa Cruz, Dallas, TX, USA) for an exosome-negative marker, and β-actin (sc-47778, Santa Cruz). After washing three times with TBST at 150 rpm for 10 min, the membranes were incubated with horseradish peroxidase-conjugated anti-mouse IgG (7076, CST) and anti-rabbit secondary antibodies (7074, CST) for 2 h. Finally, the blots were treated with enhanced chemiluminescence (Clarity Western ECL Substrate, 1705061, Bio-Rad, Hercules, CA, USA) and visualized with a Chemidoc machine (Invitrogen™ iBright™ Imagers, CL-1000, Thermo Fisher Scientific).

### 4.7. RT-qPCR

After harvesting, the cells were treated with 1 mL of Labozol (CMRZ001, Cosmogenetech, Seoul, Republic of Korea) and separated according to the manufacturer’s instructions. Total RNA was extracted from cultured cells using Labozol (CMRZ001, Cosmo Genetech). cDNA was synthesized using a Labopass^TM^ m-mulv reverse transcriptase kit(CMRT010, Cosmogenetech). Gene expression in control and exosome-treated cells was confirmed using a 2X Master Mix. cDNA was obtained from the RNA using a reverse transcriptase kit (Takara, Dalian, China), and quantitative real-time PCR was performed using SYBR Premix Ex Taq II (Takara). The cycling conditions were 25 °C for 10 min, 37 °C for 2 min, and 85 °C for 5 min. The mRNA levels were normalized to GAPDH, and 2^−ΔΔCt^ was used for data statistics. The primers used for RT-qPCR are listed in Table 1.

### 4.8. Cell Proliferation Assay

BJ cells were seeded at 3000 cells/cm^2^ in 96-well plates. After 12 h, the cells were treated with exosomes and PBS and then incubated for 24 h. Next, 5 mg/mL of MTT reagent was added, and the cells were incubated for an additional 2 h. The absorbance was measured at 540 nm using a Bio-Rad x-Mark^TM^ spectrophotometer (1681150, Bio-Rad Laboratories).

### 4.9. Transwell Migration Assay

BJ cells were seeded as 5 × 10^4^ cells in a Transwell plate (CLS3422, Corning, NY, USA). After 12 h, the exosomes and PBS were treated with 1 × 10^9^ particles/well and incubated for 24 h. Next, the cells were fixed with 4% paraformaldehyde (PFA) (P2031, Biosesang, Yongin, Korea) for 10 min, followed by treatment with 100% MeOH for 10 min. Finally, the cells were stained with a crystal violet solution (V5265, Sigma) and observed.

### 4.10. Wound Closure Assay

After seeding BJ cells in a 6-well plate, when the density reached 90%, 10 μg/mL of mitomycin C (M4287, Sigma, St. Louis, MO, USA) was added for 2 h. The confluent cells were uniformly scratched using a 1000 μL pipette tip across the wells, and dead cells were removed entirely. After switching to serum-free DMEM, the exosomes were treated with 1 × 10^9^ particles/well. The wound regions were photographed immediately and at 6 h intervals for 24 h. The closed area was measured using the TScratch software (https://github.com/cselab/TScratch, accessed on 28 September 2024) and calculated as a percentage.

### 4.11. NO Assay

After seeding the RAW cells at a density of 1 × 10^5^ cells per plate in 24-well plates, the cells were allowed to adhere for 12 h. Subsequently, the RAW cells were treated with LPS from Sigma (L4391-1MG), exosomes, and Dex (5000222, Peprotech, Waltham, MA, USA). After 24 h of treatment, 100 μL of the supernatant from each well was harvested and mixed with the reagent in a 1:1 ratio. The absorbance of the mixture was measured at 540 nm. Following measurement, all the remaining media in each well were harvested, and the cells were treated with Rabozol to obtain cell lysates. The Rabozol-treated cells were isolated to check the gene expression levels of iNOS and COX2.

### 4.12. Wound-Healing Mouse Model

Six-week-old BALB/c mice (female; 20 ± 2 g) were purchased from JA BIO (Gyeonggi-do, Republic of Korea). All the experimental procedures were approved by the Institutional Animal Care and Use Committee (IACUC) of Konkuk University (Seoul, Republic of Korea) (IACUC: KU 21234). All animals were housed for 1 week before the experiment for acclimatization in a well-ventilated room with adjusted temperature and humidity and a 12 h light/12 h dark cycle. Food and water were provided ad libitum.

Mice were anesthetized at a ratio of 4:1 with Alfaxan and Rompun. The mice were then laid down and wounded on the back with an 8 mm skin biopsy punch (P0000BFT, KAI). Wound healing was facilitated by injecting WJ MSC exosomes, MAP peptide exosomes, and FP2 peptide exosomes diluted with 20 μL of 1 × 10^9^ particles around the incision site. The wound size was observed at intervals of 0, 3, 7, and 10 d. The effect of the exosomes was confirmed by measuring the wound area using ImageJ software (v7.8).

### 4.13. Histological Analysis

After monitoring the wound areas, the mice were sacrificed. The wound areas were collected and fixed with 4% paraformaldehyde (SM-P01-100, GeneAll Biotechnology, Norcross, GA, USA). The tissues were gently washed with PBS to remove the PFA. After dehydration using graded alcohol solutions, paraffin embedding was performed. Then, 4 μm thick tissue slices were cut perpendicular to the wound surface. The cut tissues were placed on precoated slides with 0.1% *w*/*v* poly L-lysine (P8920, Sigma). Hematoxylin and eosin staining was performed to confirm the regeneration of the wound areas. Masson’s trichrome staining was also performed to assess the degree of collagen synthesis. The tissue slides were scanned using a digital slide scanner (3D Histech, Budapest, Hungary), as described in our previous report [54].

### 4.14. Statistical Analysis

The experiments were repeated at least thrice. Statistical analyses were performed using the GraphPad Prism software 9.0 (GraphPad Software, San Diego, CA, USA). The data are displayed as means ± SEM. Unpaired two-tailed Student’s *t*-tests (two groups) and one-way analysis of variance were performed to compare values and evaluate the statistical significance. Statistical significance was set at *p* < 0.05.

## 5. Conclusions

In this study, we demonstrated that the immobilization of FP2 on culture plates significantly enhances the proliferation of WJ MSCs and boosts exosome yield, showcasing a novel strategy for optimizing exosome production. FP2-exo exhibited enhanced migration effects in fibroblasts, effectively reduced the expression of pro-inflammatory factors in macrophages, and accelerated skin wound healing in vivo through the activation of the FGF2-AKT signaling pathway. While these findings underscore the therapeutic potential of FP2-exo in wound healing, further research is required to investigate additional signaling pathways that may contribute to its effects beyond the FGF2-AKT axis. Moreover, limitations such as the need to evaluate the long-term effects of FP2 on cell functionality and potential cytotoxicity remain. To address these challenges, future studies should incorporate long-term in vitro and in vivo assessments to ensure cell viability and functional integrity over time. This research not only positions FP2 as a promising tool for regenerative medicine but also sets the stage for innovative approaches to harness exosome-based therapies, opening new avenues for both basic research and clinical applications.

## 6. Patents

We filed patents for this study (Korean patent application numbers: 10-2023-0069536, 10-2023-0071436, and 10-2023-0071437).

## Figures and Tables

**Figure 1 ijms-25-10709-f001:**
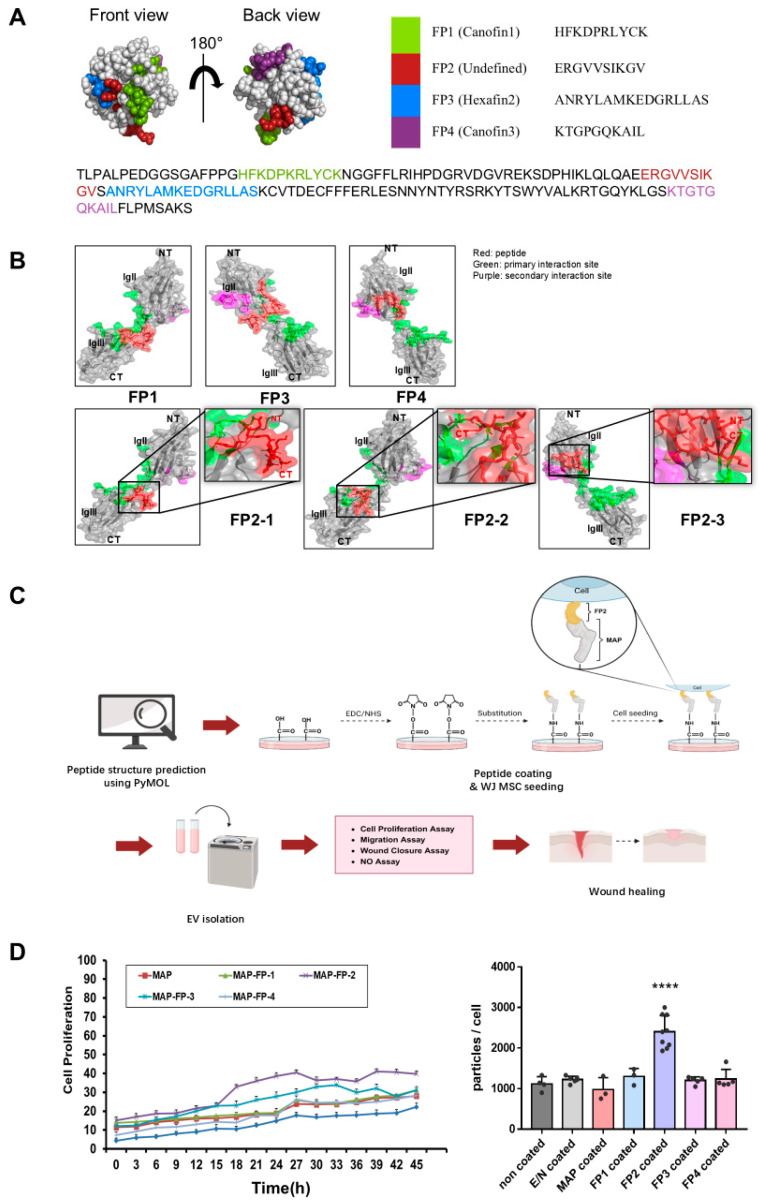
Examination of the structure and functional potential of fibroblast growth factor 2 (FGF2)-derived peptides. (**A**) Prediction of the positions of four peptides within FGF2 using PyMOL. (**B**) Structures of canofin1, FP2, hexafin2, and canofin3. Green indicates the primary interaction site, magenta indicates the secondary interaction site, and red represents the peptide. (**C**) Schematic overview of this study after screening peptides. The illustration was created using BioRender.com. (**D**) Comparison of cell proliferation and exosome production in Wharton’s jelly mesenchymal stem cells (WJ MSCs) coated with different peptides. Cell proliferation on peptide-coated plates was monitored for 45 h in an incubator. Following the medium’s change to αMEM with 10% exosome-depleted FBS, supernatants were collected after 48 h to isolate exosomes. The data are presented as the means ± standard deviation (n = 3). Results with a *p*-value < 0.05 were considered statistically significant. **** *p* < 0.0001 compared with the non-coated group.

**Figure 2 ijms-25-10709-f002:**
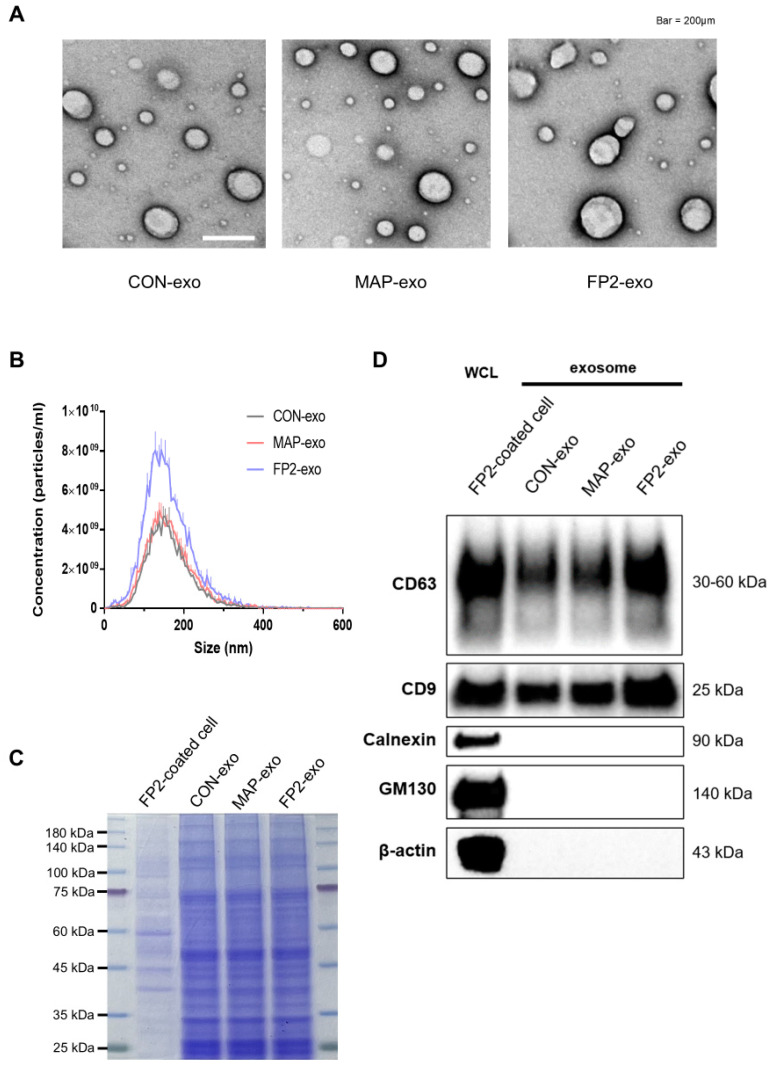
Characterization of exosomes isolated from WJ MSCs on uncoated plates (CON-exo), from WJ MSCs on plates coated with MAP (MAP-exo), and from WJ MSCs on plates coated with FP2 (FP2-exo). (**A**) Transmission electron microscopy images showing the exosome morphology of CON-exo, MAP-exo, and FP2-exo. Scale bar: 200 µm. (**B**) Nanoparticle-tracking analysis (NTA) graphs of CON-exo, MAP-exo, and FP2-exo, indicating exosome size distribution. (**C**) Coomassie blue-stained image for normalizing the exosome samples in the Western blot analysis. (**D**) Western blot images of CD63, CD9, calnexin, GM130, and β-actin expression in CON-exo, MAP-exo, FP2-exo, and WJ MSCs (WCL—whole-cell lysate).

**Figure 3 ijms-25-10709-f003:**
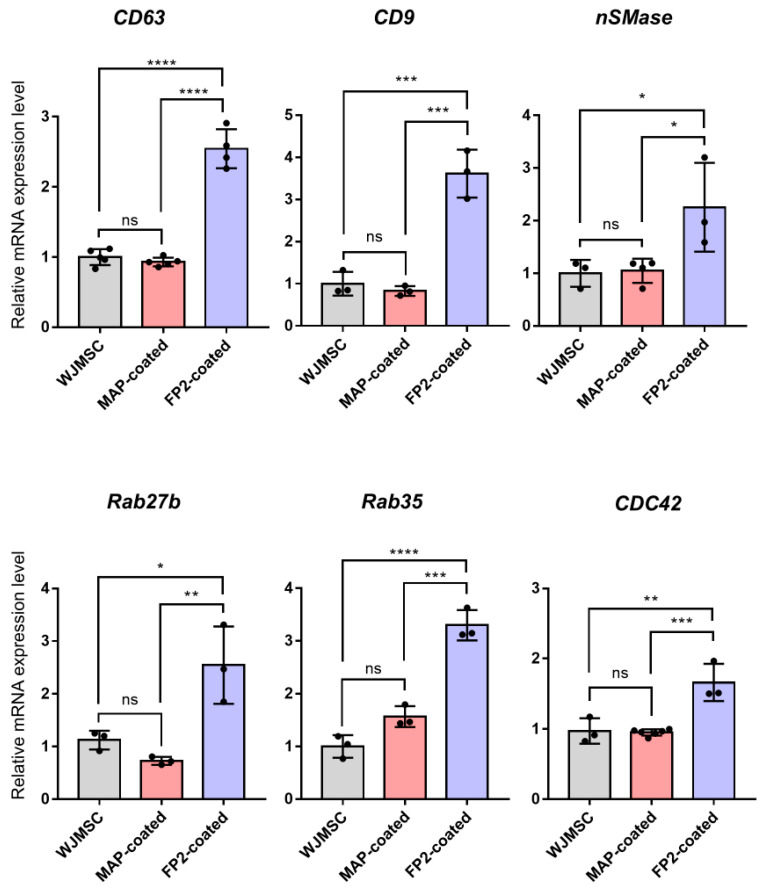
Verification of the difference in gene expression levels associated with exosome biosynthesis. Gene expression related to exosome biosynthesis in WJ MSCs cultured on uncoated plates, plates coated with MAP, and plates coated with FP2. To assess gene expression differences between WJ MSCs treated with peptides and untreated cells, samples were collected 3 d post-seeding for RNA extraction. The data are presented as means ± standard deviation (n = 3). Results with a *p*-value < 0.05 were considered statistically significant. * *p* < 0.05, ** *p* < 0.01, *** *p* < 0.001, **** *p* < 0.0001, and ns = not significant.

**Figure 4 ijms-25-10709-f004:**
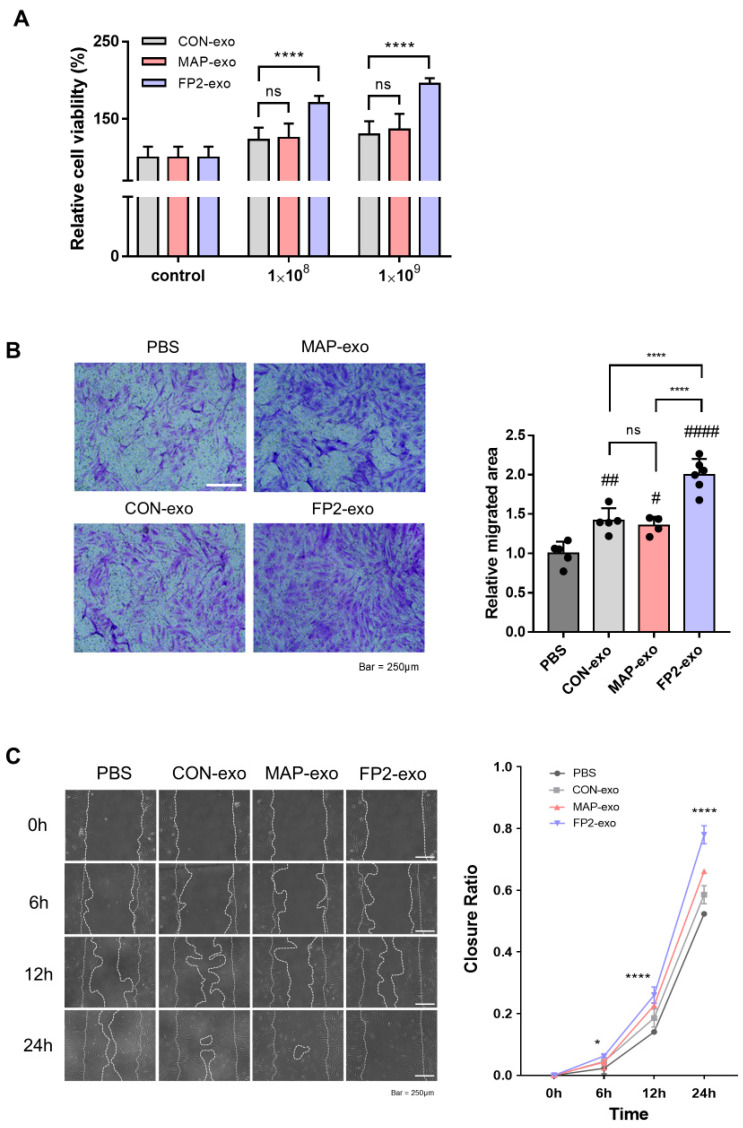
Confirmation of the migration-enhancing effect of FP2-exo in human fibroblasts. (**A**) Following treatment with exosomes in BJ cells, cell viability was confirmed. The cell viability values were normalized by setting the control values as 100%. The data are presented as means ± standard deviation (n = 3). Results with a *p*-value < 0.05 were considered statistically significant. **** *p* < 0.0001 and ns = not significant. (**B**) Images and a graph from the Transwell migration assay. BJ cells were seeded in the Transwell at 5 × 10^4^ cells/cm^2^. After 12 h, 1 × 10^9^ exosomes were applied, and the cell migration effects were assessed by staining with crystal violet after 24 h. Scale bar: 200 µm. The data are presented as means ± standard deviation (n = 3). Results with a *p*-value < 0.05 were considered statistically significant. # *p* < 0.05, ## *p* < 0.01, and #### *p* < 0.0001 compared with the PBS group. **** *p* < 0.0001 and ns = not significant. (**C**) Images and graphs from the scratch assay. BJ cells were seeded in a 6-well plate at 2.5 × 10^5^ cells/cm^2^ and incubated until they reached 100% confluency. Subsequently, the cells were treated with 10 μg/mL of mitomycin C for 2 h, followed by scratching with a 1 mL tip. Images were taken 6, 12, and 24 h after treatment with 1 × 10^9^ exosomes. Scale bar: 250 µm. The data are presented as means ± standard deviation (n = 3). Results with a *p*-value < 0.05 were considered statistically significant. * *p* < 0.05 and **** *p* < 0.0001 compared with the PBS-treated group.

**Figure 5 ijms-25-10709-f005:**
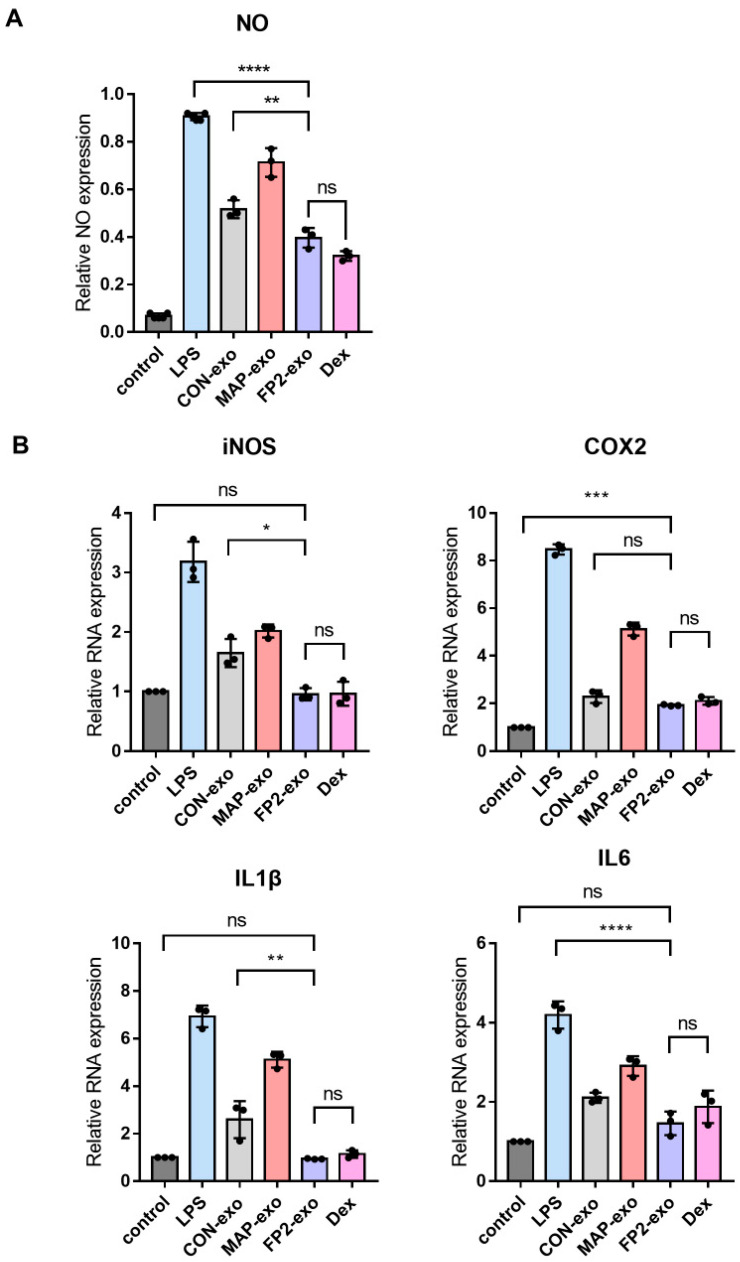
Characterization of anti-inflammation activity in the exosomes. (**A**) The presence of nitric oxide (NO) in the cell-free supernatant of RAW cells treated with LPS was assessed. (**B**) Following RNA isolation from LPS-treated RAW 264.7 cells, the expression levels of inducible iNOS, COX2, IL1β, and IL6 were determined using RT-qPCR. The data are presented as means ± standard deviation (n = 3). Results with a *p*-value < 0.05 were considered statistically significant. * *p* < 0.05, ** *p* < 0.01, *** *p* < 0.001, **** *p* < 0.0001, and ns = not significant.

**Figure 6 ijms-25-10709-f006:**
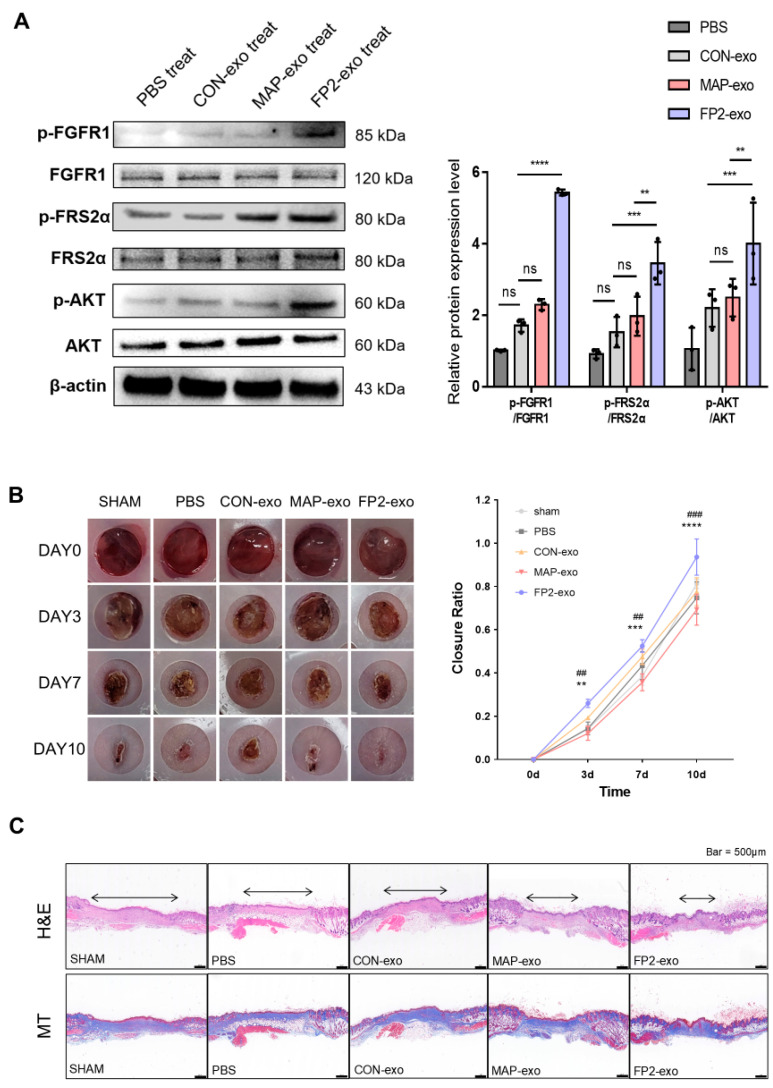
Confirming the healing of mouse wounds through exosome therapy. (**A**) Western blot image showing the FGF2 signal in exosome-treated BJ cells. Cells were treated with exosomes and harvested after 2 h to isolate proteins. The expressions of p-FGFR1, FGFR1, p-FRS2α, FRS2α, p-AKT, and AKT were observed. The data are presented as means ± standard deviation (n = 3). Results with a *p*-value < 0.05 were considered statistically significant. ** *p* < 0.01, *** *p* < 0.001, **** *p* < 0.0001, and ns = not significant. (**B**) Wound-healing progress after treated exosomes was monitored by capturing photos every 0, 3, 7, and 10 d using an 8 mm punch for wound creation. The data are presented as means ± standard deviation (n = 3). Results with a *p*-value < 0.05 were considered statistically significant. ^##^ *p* < 0.01, ^###^
*p* < 0.001, and compared with sham group. ** *p* < 0.01, *** *p* < 0.001, and **** *p* < 0.0001 compared with PBS group. (**C**) Histological analysis of the wound area via hematoxylin and eosin and Masson’s trichrome staining. Scale bar: 500 µm.

**Table 1 ijms-25-10709-t001:** Primers used for RT-qPCR.

Gene	Forward Primer Sequence (5′ to 3′)	Reverse Primer Sequence (5′ to 3′)
h*CD63*	CAA CCA CAC TGC TTC GAT CCT G	GAC TCG GTT CTT CGA CAT GGA AG
h*CD9*	TCG CCA TTG AAA TAG CTG CGG C	CGC ATA GTG GAT GGC TTT CAG C
h*CDC42*	TGA CAG ATT ACG ACC GCT GAG TT	GGA GTC TTT GGA CAG TGG TGA G
h*nSMase*	GAA GCA CAC CTC AGG ACC AAA G	CAG CCA GTC CTG AAG CAG GTC
h*Rab27b*	TAG ACT TTC GGG AAA AAC GTG TG	AGA AGC TCT GTT GAC TGG TGA
h*Rab35*	CAG CCC ATC TTA CTG CAA GCA G	GCT GAC AAC CTG TCG GAG AGA A
m*iNOS*	GAG ACA GGG AAG TCT GAA GCA C	CCA GCA GTA GTT GCT CCT CTT C
m*COX2*	CTC ACG AAG GAA CTC AGC AC	GGA TTG GAA CAG CAA GGA TTT G
m*IL1β*	TGG ACC TTC CAG GAT GAG GAC A	GTT CAT CTC GGA GCC TGT AGT G
m*IL6*	TAC CAC TTC ACA AGT CGG AGG C	CTG CAA GTG CAT CAT CGT TGT TC

## Data Availability

The data are contained within this article.

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
