# Peer review of "The Immobilization of an FGF2-Derived Peptide on Culture Plates Improves the Production and Therapeutic Potential of Extracellular Vesicles from Wharton’s Jelly Mesenchymal Stem Cells"

_ijms, 2024, doi:10.3390/ijms251910709_

Round 1

Reviewer 1 Report

Comments and Suggestions for Authors

I recommend addressing these key areas to enhance the manuscript's quality.

1. To strengthen the manuscript, I suggest the authors focus on the critical areas outlined in lines 79-82 on page 2 and provide supporting references.

2. Unclear figure labels hinder accurate referencing in the written analysis. To rectify this issue, the authors should label all figures clearly and consistently.

3. When presenting western blot results, authors should clearly label key details, such as the molecular weight markers in kilodaltons (kDa).

4. To enhance transparency and enable comprehensive evaluation, the authors should include the original, full-size raw western blot images (Fig. 2C, Fig. 6A) as supplementary materials. When presenting western blot data, it is critical that authors clearly label key details such as the samples analyzed, and the gel percentage used. Providing this vital information alongside the full-size raw images will significantly improve the clarity and reproducibility of the findings.

5. How did the authors normalize the exosome samples in the western blot analysis and then select the FP2-exo samples with the highest expression levels of the exosome markers CD9 and CD63 for further consideration?

6. The manuscript should include high-magnification images of the same tissue areas from each experimental group at each time point, displaying both hematoxylin and eosin (H&E) staining and immunohistochemical (IHC) staining for wound healing and inflammatory markers.

7. To clearly identify the specific cell types expressing the protein of interest, the authors should include arrows in the IHC figures.

8. The authors must clearly outline the key strengths and limitations of their research, and then propose innovative directions for future investigation in this field.

Comments on the Quality of English Language

Rephrasing problematic sentences can enhance the clarity and cohesion of the text. Consulting a native English speaker or professional editing service is strongly recommended to ensure the narrative is polished and flows seamlessly.

Author Response

1. Summary

Thank you very much for taking the time to review this manuscript. Please find the detailed responses below and the corresponding revisions/corrections highlighted/in track changes in the re-submitted files. (The revisions have been highlighted in blue text within the manuscript.)

2. Point-by-point response to Comments and Suggestions for Authors

Comments 1: To strengthen the manuscript, I suggest the authors focus on the critical areas outlined in lines 79-82 on page 2 and provide supporting references.

Response 1: We appreciate your suggestion. We have included additional supporting references to strengthen our manuscript.

Comments 2: Unclear figure labels hinder accurate referencing in the written analysis. To rectify this issue, the authors should label all figures clearly and consistently.

Response 2: Thank you for this observation. We have revised all figures to ensure they are clearly labeled and consistent throughout the manuscript, facilitating accurate referencing.

Comments 3: When presenting western blot results, authors should clearly label key details, such as the molecular weight markers in kilodaltons (kDa).

Response 3: We have addressed this comment by clearly labeling the molecular weight markers in kilodaltons (kDa) in all western blot figures to enhance clarity and understanding.

Comments 4: To enhance transparency and enable comprehensive evaluation, the authors should include the original, full-size raw western blot images (Fig. 2C, Fig. 6A) as supplementary materials. When presenting western blot data, it is critical that authors clearly label key details such as the samples analyzed, and the gel percentage used. Providing this vital information alongside the full-size raw images will significantly improve the clarity and reproducibility of the findings.

Response 4: Thank you for your valuable feedback regarding the inclusion of original, full-size raw western blot images and labeling details.

In response to your suggestion, we have included all full-size raw western blot images as supplementary materials (Non-published Material). Each image clearly displays the molecular weight markers in kilodaltons, ensuring that the data is easily interpretable.

The methodology for our western blot analysis, including the samples analyzed and the gel percentage used, is thoroughly described in the Materials and Methods section of our manuscript, providing additional clarity on our experimental procedures.

Furthermore, we recognized that the existing western blot image for Figure 6A was based on older data with lower resolution. To improve the quality and clarity of our results, we conducted new western blot experiments to validate reproducibility. This allowed us to generate high-resolution images that better represent our findings, and we have revised the original figure accordingly.

We appreciate your insightful comments and believe that these revisions will significantly enhance the clarity and reproducibility of our findings.

Comments 5: How did the authors normalize the exosome samples in the western blot analysis and then select the FP2-exo samples with the highest expression levels of the exosome markers CD9 and CD63 for further consideration?

Response 5: Thank you for your insightful question regarding the normalization of exosome samples in the western blot analysis and the selection process for FP2-exo samples.

To normalize the exosome samples in our western blot analysis, we performed Coomassie blue staining of the gel. This method allowed us to quantify the total protein loaded in each lane, ensuring that equal amounts of protein were compared across samples. Each sample was loaded at 10 μg, and the quantification was conducted using a BCA kit, as described in our Materials and Methods section.

Regarding the selection of FP2-exo samples, it is important to clarify that we did not select these samples based solely on western blot results. Instead, we utilized computational tools, specifically PyMOL and CABS-dock, to screen for the most effective peptide prior to exosome analysis. The western blot analysis, along with transmission electron microscopy (TEM) and nanoparticle tracking analysis (NTA), was performed to confirm that the FP2 coating did not alter the essential characteristics of the exosomes. This comprehensive evaluation ensured that the exosome properties remained intact, and our findings indicated that FP2-exo exhibited the highest yield without compromising exosome marker expression.

We appreciate your feedback and hope this response clarifies our methodology and selection criteria.

Comments 6: The manuscript should include high-magnification images of the same tissue areas from each experimental group at each time point, displaying both hematoxylin and eosin (H&E) staining and immunohistochemical (IHC) staining for wound healing and inflammatory markers.

Response 6: Thank you for your valuable suggestion.

In terms of the methodology, we performed precise in vivo experiments by creating wounds of uniform size using an 8-mm punch. To ensure consistency in our photographic records, we placed an 8 mm plastic ring over the wound site during imaging to maintain the same magnification across all time points. This approach helped standardize our assessment of the wound healing process. We are committed to providing the most accurate and informative data possible.

As we have high-magnification H&E images for the relevant tissue areas, we have incorporated them into the revised manuscript. As for immunohistochemical (IHC) staining, while we acknowledge its importance for further validating wound healing and inflammation markers, we currently do not have this data for the study presented. Instead, we have included Masson's trichrome staining results to evaluate collagen synthesis and the degree of wound healing, which also provides crucial insights into tissue regeneration. We fully intend to perform IHC staining in subsequent studies to strengthen the current findings and elucidate the expression profiles of wound healing and inflammatory markers.

To summarize, by incorporating Masson's trichrome staining and high-magnification H&E images, we have provided substantial histological evidence supporting the regenerative effects of FP2-exo. Future studies will include additional staining techniques, such as IHC, to further build upon this work. We have updated the Discussion section accordingly to reflect these revisions, which are emphasized in blue. We hope these revisions and our future plans address your concerns.

Comments 7: To clearly identify the specific cell types expressing the protein of interest, the authors should include arrows in the IHC figures.

Response 7: We appreciate this suggestion regarding the clear identification of cell types in the IHC figures. As mentioned, we currently do not have IHC data in the present study, but we have included Masson's trichrome staining results, which offer valuable insight into collagen deposition and overall tissue structure. In future investigations, we plan to include IHC analysis and will ensure that we follow your recommendation by using arrows to clearly indicate the specific cell types expressing the proteins of interest.

Comments 8: The authors must clearly outline the key strengths and limitations of their research, and then propose innovative directions for future investigation in this field.

Response 8: Thank you for this insightful comment. In response, we have expanded the conclusion section of our manuscript to clearly address both the strengths and limitations of our research, along with proposing future directions for investigation.

We emphasize the novelty of our approach, particularly the immobilization of FP2 onto culture plates, which significantly enhances the proliferation of WJ MSCs and increases exosome yield. FP2-exo demonstrated therapeutic potential by enhancing fibroblast migration, reducing pro-inflammatory factors in macrophages, and accelerating skin wound healing through activation of the FGF2-AKT signaling pathway. These strengths highlight FP2-exo as a promising tool for regenerative medicine.

However, we acknowledge limitations, including the need for long-term studies to assess potential cytotoxicity and the effects of FP2 on cell functionality. We propose that future research should explore additional signaling pathways contributing to FP2-exo’s therapeutic effects and include long-term in vitro and in vivo studies to ensure cell viability and functional integrity.

We hope this revised conclusion satisfactorily addresses your concerns while suggesting innovative directions for future exploration.

Reviewer 2 Report

Comments and Suggestions for Authors

The manuscript entitled "Immobilization of an FGF2-Derived Peptide on Culture Plates Improves the Production and Therapeutic Potential of Extracellular Vesicles from Wharton's Jelly Mesenchymal Stem Cells" addresses an interesting topic and proposes a study on obtaining exosomes for wound healing using a fragment of peptides derived from fibroblast growth factor 2 (FP2). The presented experimental results show that following the coating of Wharton mesenchymal stem cells (WJ MSC) with FP2, there is an increase in cell proliferation. Basically, fibroblasts treated with exosomes from WJ MSCs showed an increase in FGF2 expression and a wound healing effect demonstrated in an in vivo experiment.

The work is well organized, the experiments are well described and the results are clearly presented, however some small clarifications are needed.

I recommend publishing this article after a minor revision.

I suggest the following to the authors:

1. Figure 1 should be brought to a better resolution, or even presented as two distinctive figures, to be easier to understand. In addition, A should be written in the legend of the figure); B) to correlate with the data in the figure. Likewise, in the legends of the other figures.

2. It should be specified what the term AKT represents where it appears the first time.

3. After the discussion chapter, a conclusion chapter could be added. Successful experiments on mice indicated a clear healing of wounds, as a result of the effect of the injected exosomes. A few well-pointed conclusions could increase readers' interest in this article.

Author Response

1. Summary

Thank you very much for taking the time to review this manuscript. Please find the detailed responses below and the corresponding revisions/corrections highlighted/in track changes in the re-submitted files. (The revisions have been highlighted in blue text within the manuscript.)

2. Point-by-point response to Comments and Suggestions for Authors

Comments 1: Figure 1 should be brought to a better resolution, or even presented as two distinctive figures, to be easier to understand. In addition, A should be written in the legend of the figure); B) to correlate with the data in the figure. Likewise, in the legends of the other figures.

Response 1: We have improved Figure 1 by enhancing its resolution and reordering the elements for greater clarity. Additionally, we have revised the figure legend, correlating each section with appropriate labels (e.g., 'A,' 'B') to ensure the data aligns accurately with the legend.

Comments 2: It should be specified what the term AKT represents where it appears the first time.

Response 2: Thank you for pointing this out. In response to your suggestion, we have revised the manuscript to specify that AKT refers to 'Protein Kinase B' when it is first mentioned. We believe this clarification will provide readers with a better understanding of our study. This revision has been highlighted in blue text on pages 7, lines 157-158.

Comments 3: After the discussion chapter, a conclusion chapter could be added. Successful experiments on mice indicated a clear healing of wounds, as a result of the effect of the injected exosomes. A few well-pointed conclusions could increase readers' interest in this article.

Response 3: Thank you for your valuable suggestion. In response, we have added a conclusion section to the manuscript to succinctly summarize our key findings and emphasize the successful outcomes of our wound healing experiments using FP2-exo. The revised conclusion highlights the therapeutic potential of FP2-exo in accelerating wound healing through the activation of the FGF2-AKT signaling pathway. We believe this addition will provide a clear and impactful summary of the research, increasing reader engagement and interest in the article, as suggested.

Round 2

Reviewer 1 Report

Comments and Suggestions for Authors

In the revised article, the authors comprehensively addressed the reviewer's comments by modifying the manuscript and figures, and providing detailed point-by-point responses.